# Investigating the Balance between Structural Conservation and Functional Flexibility in Photosystem I

**DOI:** 10.3390/ijms25105073

**Published:** 2024-05-07

**Authors:** Nathan Nelson

**Affiliations:** Department of Biochemistry and Molecular Biology, The George S. Wise Faculty of Life Sciences, Tel Aviv University, Tel Aviv 69978, Israel; nelson@tauex.tau.ac.il

**Keywords:** photosynthesis, photosystem I (PSI), electron transfer, light harvestin

## Abstract

Photosynthesis, as the primary source of energy for all life forms, plays a crucial role in maintaining the global balance of energy, entropy, and enthalpy in living organisms. Among its various building blocks, photosystem I (PSI) is responsible for light-driven electron transfer, crucial for generating cellular reducing power. PSI acts as a light-driven plastocyanin-ferredoxin oxidoreductase and is situated in the thylakoid membranes of cyanobacteria and the chloroplasts of eukaryotic photosynthetic organisms. Comprehending the structure and function of the photosynthetic machinery is essential for understanding its mode of action. New insights are offered into the structure and function of PSI and its associated light-harvesting proteins, with a specific focus on the remarkable structural conservation of the core complex and high plasticity of the peripheral light-harvesting complexes.

## 1. Introduction

Oxygenic photosynthesis is the primary process of converting sunlight into chemical energy on Earth. Existing for nearly four billion years, it has played a crucial role in supporting advanced life on Earth. Throughout most of this period, it has been a key contributor to global entropy and a major source of total enthalpy in living organisms. The conversion of solar energy into chemical energy through photosynthesis is facilitated by two multisubunit membrane protein complexes: photosystem I (PSI) and photosystem II (PSII). Light is absorbed by pigment cofactors and the excitation energy is efficiently transferred among the antenna pigments, ultimately being converted into chemical energy.

Photosynthesis encompasses the widest spectrum of redox potential within the realm of life’s biochemistry. Navigating through extreme redox potential presents a notable challenge, primarily in safeguarding against damage caused by highly reactive elements like singlet oxygen, a by-product of photochemical activity, and by preventing the loss of high-energy electrons to unproductive elements in the surroundings. Over the course of extensive evolution, oxygenic photosynthesis has effectively addressed these challenges, as reflected in the intricate structure of its components. The determination of the atomic structure of photosynthetic complexes in diverse organisms thriving in various environments offers insight into how each component adeptly tackles the specific challenges it encounters.

### 1.1. The Molecular Architecture of a Primordial PSI Core Complex

The evolution of PSI was likely initiated by the formation of a homodimeric reaction center, similar to the one currently present in green bacteria [1]. Gene duplication gave rise to a heterodimeric reaction center that subsequently evolved into the PSI found in cyanobacteria, algae, and plant chloroplasts [2]. Arguably, the heterodimeric structure evolved to induce excitation slippage rather than to increase efficiency [3]. The high-resolution structure of homomeric PSI complexes was recently elucidated [4,5,6]. Comparing the basic structure of homomeric PSI in green bacteria with that in cyanobacteria and higher plants reveals both fundamental conservation and an evolutionary quantum leap. Aligning the *Chlorobium* sequence with the cyanobacterial or higher plant sequences revealed a notably low identity of approximately 30% over 80 amino acids. However, green sulfur bacterial RC contains only 30 chlorophylls and cyanobacterial and higher plant PSI contain 87–88 Chls. In other words, only 23 out of 87–88 Chls are conserved between green sulfur bacterial RC and PSI [7]. In contrast, a striking structural similarity across the entire subunits was observed in cyanobacteria, algae, and plants (Figure 1a). Additionally, 23 chlorophyll molecules were found at almost identical locations (Figure 1b). The evolution of this phenomenon is not readily apparent. How can structural identity evolve in the absence of amino acid sequence homology? An evolutionary quantum leap occurred in the development of cyanobacterial PSI. Organisms that bridge the gap between homomeric and heterodimeric PSI have not yet been discovered. The PSI reaction centers of cyanobacteria and plants are highly homologous to each other. It is amazing that, despite nearly two billion years of evolutionary divergence and enormous differences in habits, in terms of ecological niches (light intensity and temperature), the photosynthetic apparatus of higher plants has remained very similar to that of cyanobacteria.

Figure 2 depicts the structural homology between Synechocystis and pea plants PSI core subunits. They share approximately 80% amino acid sequence identity and exhibit significant structural similarity. It is remarkable that 84 out of 88 chlorophyll molecules maintain almost identical positions in the core subunits PsaA and PsaB. There is no explanation of how such an arrangement can be maintained during the last two billion years, a period in which primordial eukaryotes engulfed cyanobacteria to generate chloroplasts. The mechanism by which numerous random mutations that might even slightly change the chlorophyll position were genetically reported and corrected remains unclear.

Even more puzzling is the short evolutionary period during which a random sequence of about 700 amino acids evolved to form PsaA and PsaB, presumably occurring over a few hundred million years when the Earth’s surface became habitable for life and the completion of PSI core architecture took place. It is proposed that it should have taken the entire 4 billion years for the evolution of primordial organisms to the current life systems. A similar argument could be relevant to other highly complex systems such as RNA and DNA replicases, ribosomes, ATP-synthase, and other intricate biochemical assemblies. Therefore, it prompts me to consider local panspermia for the early initiation of life on Earth. In contrast to the general assumption that life spreads all over the universe, local panspermia refers to unique and single opportunistic events that enable life on Earth. It considers the Fermi paradox and the extensive discussion that it generates [8,9,10,11]. Accordingly, a star positioned in the current location of the sun within the Milky Way once hosted an Earth-like terrestrial planet, possessing most of the characteristic features that endured for over 5 billion years. This celestial body eventually experienced a supernova explosion, dispersing its oceans into space. Some remnants of these oceans were transported as meteorites containing primitive life forms, contributing to local panspermia and initiating life on Earth. This process includes the universal genetic code that encodes the primordial complex systems mentioned above. Consequently, if we represent the Drake equation as N = r, where r is exceptionally rare (as low as 10^−12^), the equation now transforms into N = r^2^. This implies that life on Earth is not only unique in our galaxy but is also practically unique in the entire universe.

### 1.2. Building Minimal Functional PSI

Photosystem I in higher plants is characterized as a light-induced plastocyanin-ferredoxin oxido-reductase [12]. Furthermore, in cyanobacteria and algae, alternative electron donors and acceptors are present [13,14,15,16,17,18,19,20,21,22,23,24,25,26,27,28,29,30,31]. Beginning with cyanobacteria, each PSI complex contains PsaC, a conserved two-iron-sulfur protein denoted as FA and FB. PsaC plays a crucial role in stabilizing the very low reduction potential resulting from light-induced excited electrons, thereby safeguarding the core complex against the formation of damaging radicals. This assembly exhibits inefficiency in donating and accepting electrons. It has been documented that the favored excitation pathway in PSI of plants and green algae is Branch B [32,33,34]. Therefore, we assign the excitation pathway related to slippage to Branch A [3]. This proposal is substantiated by the variation in the position of the isoprenoid side chain of the quinone in Branch A observed across different organisms and even within the same species [35,36]. The structural analysis of PSI complexes clearly indicates that Branch A is more exposed to the membrane environment compared to Branch B [37]. Additionally, the arrangement of the phytol chains and surrounding amino acids around the quinone in Branch B is more conserved than in Branch A. This circumstance strongly implies that Branch B evolved to optimize coupling, while Branch A evolved to facilitate protective slippage. Consequently, throughout the course of evolution, additional subunits have emerged to augment the oxidation–reduction process.

Recently, a PSI assembly intermediate present during the greening of etiolated oat leaves was isolated [36]. The high-resolution cryo-EM structure, which is named pre-PSI-1, revealed the presence of eight subunits: PsaA, PsaB, PsaC, PsaD, PsaE, PsaH, PsaI, and PsaL. The structure suggests a complete assembly of the ferredoxin binding site of PsaD and PsaE, presumably occurring right after PsaC, along with additional functions provided by PsaH, PsaI, and PsaL. The assembly intermediate bears a resemblance to PSI encoded by marine viruses [38]. The absence of PsaF prevents efficient plastocyanin binding and, consequently, hinders efficient NADP photoreduction activity. Further assembly of PsaF, PsaK, and PsaG (in eukaryotes) completes the assembly of the basic minimal PSI that, with slight variations, is present in all photosynthetic organisms from cyanobacteria to higher plants. Figure 3 illustrates the structure of common subunits found in the photosystem I (PSI) of cyanobacteria, algae, and higher plants. Chlorophyll a serves as the universal pigment, playing a crucial role in light absorption and the transfer of excitation to the P700 reaction center. An exception is the cyanobacteria *A. Marina*, where two chlorophyll d and two pheophytins substitute chlorophyll a in the reaction center, as well as the far-red absorption chlorophyll d in the periphery [39,40].

### 1.3. Variations in the Cyanobacteria

The main strategy for mitigating light and environmental stresses entails integrating light-harvesting peripheral proteins. Cyanobacteria play a major role in Earth’s most critical environmental challenge—the oxygenation of the atmosphere. This phenomenon resulted in the oxidation of soluble iron in the oceans, causing it to become insoluble. Subsequently, the oxidized iron settled, restricting its accessibility to photosynthetic organisms [3,41]. The establishment of a ring of light-harvesting IsiA around PSI represents one of several metabolic adaptations designed to overcome the iron limitation imposed by the emergence of oxygenic photosynthesis [42,43]. Additionally, like any other light-harvesting antenna, it broadens the absorption landscape and enables effective photosynthesis at limited light intensities. The role of IsiA in the adaptation of cyanobacteria to stress is not limited to low-iron conditions [44]. IsiA is induced and is required for growth in multiple stress conditions, including high-light and oxidative stress. Similar strategies are used by several other cyanobacteria that must adapt to restricted environments.

The initiation of a significant evolutionary advancement began with the emergence of specialized light-harvesting complexes linked to PSI. These complexes evolved gradually, driven by the necessity to adapt to the environment and optimize light harvesting efficiency during photosynthesis. It started with the emergence of a cyanobacterial single-helix protein resembling part of algal and plant light-harvesting complexes (LHCs) [45]. It was proposed that the original function of LHCs was not to collect light and to transfer its energy content to the reaction centers but to disperse the absorbed energy of light in the form of heat or fluorescence like the non-photochemical quenching role recently shown for higher plant psbS proteins [46].

### 1.4. Red Algae PSI-LHC Supercomplexes

The fundamental components of PSI remain largely conserved throughout evolution, with some core subunits being either recruited or lost. Among the 12 PSI core subunits, nine (PsaA, B, C, D, E, F, I, J, and L) are conserved from cyanobacteria to higher plants. Additionally, the oligomerization state of the PSI core undergoes changes across prokaryotic and eukaryotic organisms [47]. In contrast, significant variations are observed in the LHC proteins and the pigments they bind. Cyanobacterial PSI lacks transmembrane LHC and may be associated with hydrophilic phycobilisomes as its antenna [48]. Conversely, the eukaryotic organism PSI features membrane-spanning LHCs as their antennas. The light-harvesting complex I (LHCI) in eukaryotic organisms exhibits notable differences in sequences, the number bound to each PSI core, and the associated pigments. Red algal PSI possesses either three or five red algal-type LHC (LHCR) antenna subunits with chlorophyll (Chl a) and zeaxanthins as the primary light-harvesting pigments [49,50]. While LHCI in both green algae and higher plants within the green lineage binds Chl a/b and carotenoids, there is variation in the number of LHCI subunits attached to each PSI core [51,52,53,54,55]. In the red lineage, numerous eukaryotic algae originating from secondary endosymbiosis exhibit unique LHCs (FCPs) that bind Chl a/c and fucoxanthins [55,56,57,58]. Within the core complex, the small membrane-spanning subunit PsaM, found in cyanobacteria and red algae but absent in green algae and higher plants, is present in the diatom PSI. The diatom PSI lacks PsaG, H, K, N, O, and X subunits. PsaX is exclusive to cyanobacterial PSI, whereas PsaG, H, and N are exclusive to green algae and higher plants. PsaK is present in all oxyphototrophs and PsaO is found in all eukaryotic oxyphototrophs, but these two subunits are absent in the diatom PSI core [59]. These features may arise from the evolutionary origin of diatoms resulting from secondary endosymbiosis with red algae. The fundamental LHCI light-harvesting complex in red algae comprises three antenna subunits with chlorophyll a and zeaxanthins as the main light-harvesting pigments [49,50]. Its placement in relation to the core complex is likely favored among all available sites; hence, it is retained in all PSI-LHC supercomplexes. However, in green algae and plants, its position was shifted and thereafter, it is consistently maintained in this new arrangement [60]. Red algae also contain two additional LHCs bound in the vicinity of PsaB and PsaL. This position is frequently utilized in several algal species but not in higher plants.

The flexibility in incorporating numerous LHC complexes into PSI in the red lineage is underscored by the structure of the extensive diatom PSI-FCPI supercomplex from the diatom *Chaetoceros gracilis* [58,59]. Figure 4 illustrates the presence of 12 core subunits and 24 fucoxanthin-chlorophyll a/c proteins (FCPIs) antenna proteins. The arrangement of 326 chlorophylls a and 34 chlorophylls c molecules enables efficient excitation transfer among them and the core complex. The 24 FCPIs assemble into an asymmetric belt surrounding the PSI core and no dimeric or tetrameric FCPIs are found (Figure 4). As diatoms stem from the secondary symbiosis of red algae, five FCPI subunits in the innermost layer remain in similar positions as Lhcrs in the red algal. The binding of the other six FCPI subunits in the innermost layer is made possible by changes in the PSI core subunits, including the substitution of PsaO by FCPI. The remaining FCPIs are probably assembled in groups in a stepwise fashion, culminating in a space-filling module. Recently, a similar structure of PSI supercomplex was reported for red tidal and coral symbiotic photosynthetic dinoflagellates [61].

### 1.5. PSI-LHC Supercomplexes of Green Algae

Corrected version: LHCI from green algae and higher plants belong to a related group of integral chlorophyll and carotenoid binding proteins with conserved pigment-binding sites and three distinct regions of transmembrane helices, featuring largely similar protein folds [62]. In green algae, the number of LHCI subunits ranges from 4 to 10 and may change in response to light and environmental stresses [63]. The green alga *D. salina* is a unicellular organism that is unique in its ability to adapt to hypersaline environments and light stress [64,65]. For this reason, it is a model organism for studying acclimation in response to abiotic environmental stresses such as high salinity, high light, and temperature [66].

The structure of a minimal PSI supercomplex was determined by crystal structure and the same preparation was analyzed using cryo-EM [15]. Only four LHCI units are present, positioned at similar positions as in the plant PSI supercomplex [67]. Further studies revealed a larger PSI supercomplex containing six LHCI functions alongside the mini complex and it was suggested that the PSI shells subunits act in response to high-light stress [34,60]. The low-light green algae *Chlamydomonas reinhardtii* and *Bryopsis corticulans* contain 10 LHCI subunits where a second half-moon belt of four LHCI is added [53,54,68]. Except for PsaO, which is necessary for efficient state transition, all the subunits present in the large Dunaliella supercomplex were also present in the above structures.

The photosynthesis state transition is a regulatory mechanism observed in plants and algae, involving the dynamic redistribution of light-absorbing complexes between photosystem I (PSI) and photosystem II (PSII). When exposed to excess light, the state transition prompts the migration of Light Harvesting Complex II (LHCII) from PSII to PSI, thereby enhancing light absorption by PSI and diminishing absorption by PSII. Conversely, in low light conditions, LHCII migrates back to PSII, ensuring a balanced excitation between the two photosystems.

During state transitions, the trimeric LHCII undergoes reversible phosphorylation and dephosphorylation, a process regulated by the redox state of plastoquinone (PQ) and controlled by a chloroplast kinase (STN7) and phosphatase (PPH1 or TAP38) in plants [69,70].

In State 1, LHCIIs are primarily associated with PSII. In response to high light and PSII overexcitation, the activation of the STN7 kinase leads to the phosphorylation of the N-terminal region of LHCII, forming the PSI-LHCI-LHCII supercomplex and resulting in a transition from State 1 to State 2. The cryo-EM analysis of this supercomplex revealed the presence of subunits PsaN and PsaO. In the large *Dunaliella* structure, PsaO and its three coordinated chlorophyll molecules were evident [60]. The structure of the large Dunaliella PSI supercomplex unveiled all potential excitation pathways between the six LHCs and the core complex. Additionally, it suggested a potential novel efficient excitation transfer between LHCII and PSI at the chlorophyll luminal leaflet. The chlorophyll positions, modeled within LHCII, revealed a novel chlorophyll binding site crucial for efficient excitation transfer from LHCII to the core complex.

Recently, high-resolution structures of the PSI–LHCI–LHCII supercomplex from *Chlamydomonas reinhardtii* have been elucidated, shedding light on the assembly mechanism between the PSI–LHCI complex and two LHCII trimers [71,72,73]. In this arrangement, one of the LHCII trimers directly interacts with PSI, while the other engages with one of the LHCIs (Lhca2). The spatial distribution of chlorophyll a and chlorophyll b molecules at the junction between the bound LHCIIs and the core PSI was measured, suggesting pathways for excitation transfer. However, upon superimposing the two core complexes, it became evident that the position of LHCII in the two preparations was significantly different (not shown). These observations clearly indicate that during the state 2 transition, the primary function of LHCII movement is to decrease the excitation landscape of PSII, rather than enhancing the excitation of PSI.

Excessive activation of photosystem I (PSI) can lead to the generation of reactive oxygen species (ROS), resulting in damage to cellular components and a compromise in the overall efficiency of photosynthesis. Therefore, it is crucial to regulate the complex, especially during the transition to state 2 with an additional antenna. This is particularly relevant in green algae, where the antenna size can reach a significant light absorption capacity. Two primary strategies exist to accomplish this objective: the first involves reducing excitation transfer from LHCII to the core complex and the second entails decreasing the number of attached LHCI. Structural information of PSI isolated from low and high light-adapted cells enlightens these aspects. In the salt-tolerant *Dunaliella* salina, the second half-moon belt is missing, resulting in six maximal functional LHCIs [60]. The presence of PsaO suggests a retention to attain the state 2 transition. The green alga *Chlorella ohadii* copes with harsh conditions including extremely high daytime illumination of ~2000 μE. Unlike other photosynthetic organisms, *C. ohadii* does not undergo photodamage, even when exposed to light intensities that are up to fourfold higher than that required to saturate the CO_2_ fixation [74]. To assess whether structural alterations in the subunit composition occurred in the PSI-HL complex, its structure was compared with that of PSI-LL [34]. The most evident differences were observed in the subunits related to the attachment of LHCII. The subunits PsaO and PsaH are known as the binding subunits of LHCII when additional light energy is required for PSI state transition [60,61]. The PSI-HL structure completely lacked the PsaO subunit, suggesting that the state transition, where LHCII moves from PSII to PSI, was not induced in *C. ohadii* under excessive HL conditions. Another smaller conformer was solved, in which the LHCI dimer, PsaO, and PsaH were entirely missing (Figure 5). A similar phenomenon was observed in the prevention of PSI overexcitation in *C. reinhardtii* [75]. Figure 6 shows a comparison between the subunit structure of optimally grown cells versus b6/f temperature-sensitive cells that were stressed at elevated temperatures. The structure of isolated PSI was very similar to the high light-grown cells. Thus, the interplay between preventing efficient excitation transfer from LHCII and reducing the number of bound LHCI is the main defense mechanism against overexcitation of PSI.

### 1.6. Interaction between PSI and Electron Donors and Acceptors

PSI acts as the mediator for NADP+ photoreduction and is specifically identified as plastocyanin-ferredoxin oxidoreductase. Although high-resolution structures for both photosystem PSI and PSII have been revealed through X-ray crystallography, attempts to crystallize membrane complexes involving donor and acceptor proteins have largely been unsuccessful. Efforts to crystallize PSI-ferredoxin achieved partial success, resulting in the development of a low-resolution crystal structure that asymmetrically binds ferredoxin to a trimeric PSI derived from the thermophilic cyanobacterium *Thermosynechococcus elongatus* [76]. Recently, the structure of ferredoxin in complex with PSI was determined at near-atomic resolution by the cryo-EM technique [77].

The cryo-EM technique proves advantageous in resolving such supercomplexes, requiring only a moderate affinity in the micromolar range to attain high-resolution structures [78]. The binding of ferredoxin is facilitated by charged amino acids, with hydrophobic interactions playing a negligible role [77]. In contrast, the binding of plastocyanin is predominantly hydrophobic and is facilitated by a flat plastocyanin structure atop the copper molecule, along with a flat hydrophobic structure that includes a sandwiched tryptophan residue from PsaA and PsaB [79].

Kinetic data for P700+ reduction by plastocyanin were analyzed using flash photolysis, revealing two phases: a fast microsecond intra-molecular electron transfer between the bound donor and P700+, followed by a subsequent slower phase where the remaining PSI complexes are reduced by the soluble donor in a bimolecular reaction with second-order kinetics. It has been observed that the dissociation constant for oxidized plastocyanin is approximately six times larger than that observed for reduced plastocyanin [80]. This could be explained by assuming conformational changes induced by P700 oxidation and reduction. Upon oxidation of plastocyanin, a slight tilt of the bound oxidized molecule allows water molecules to fill the space between plastocyanin and PSI [79]. Figure 7 illustrates the structural alterations necessary for such a mechanism.

### 1.7. The Structure of Large Supercomplexes of PSI with other Photosynthetic Membrane Complexes

The interaction among complexes within photosynthetic membranes extends beyond state transitions alone. Due to the densely packed nature of these membranes, temporary interactions are likely to occur at various stages, including development, steady-state, and stress conditions. Cryo-electron microscopy (cryo-EM) proves to be a valuable tool for elucidating the structures of large assemblies, provided they can be isolated in their native state [81,82,83,84,85,86,87,88,89]. However, the lack of high resolution in certain cases may result in published assemblies that are premature and potentially misleading. Two types of PSI complexes with different antenna sizes have been reported from the moss *Physcomitrella patens*. However, neither structure was resolved at a sufficiently high resolution to unveil detailed information about antenna organization and pigment arrangement [90]. Subsequent high-resolution studies have shown that PSI in *Physcomitrella patens* closely resembles plant PSI [91].

The recent availability of the structures of individual membrane complexes has tempted some researchers to explore interactions among these complexes through in-silico approaches or creative imagination. A more careful approach involves the use of robust biochemistry and cryo-electron microscopy (cryo-EM), utilizing cryogenic electron tomography and in situ single-particle analysis [92,93]. However, the identification and structural elucidation of large assemblies must be complemented by an investigation into their specific roles in the overall process.

Ever since the groundbreaking discovery of plant in vivo cyclic phosphorylation by Forti and Parisi in 1963 [94], the architecture of photosynthetic membrane complexes has undergone rigorous investigation. The predominant method involved isolating large membrane patches in a solution, followed by the identification of complexes through native gels and individual subunits via Western blots. This approach led to a considerable number of publications, some of which required retractions.

Recently, two landmark publications pushed the boundaries to unveil the structure of the chloroplast PSI–NDH supercomplex and discuss its role in cyclic electron transport [95,96]. The revealed structures show that PSI–NDH consists of two copies of the PSI subcomplex and one NDH complex (see Figure 8). The binding of two PSI complexes to NDH is mediated by two monomeric LHCI proteins, Lhca5 and Lhca6. Further investigations may reveal the significance of this specific structure and its contribution to plant productivity and/or stress tolerance.

## Figures and Tables

**Figure 1 ijms-25-05073-f001:**
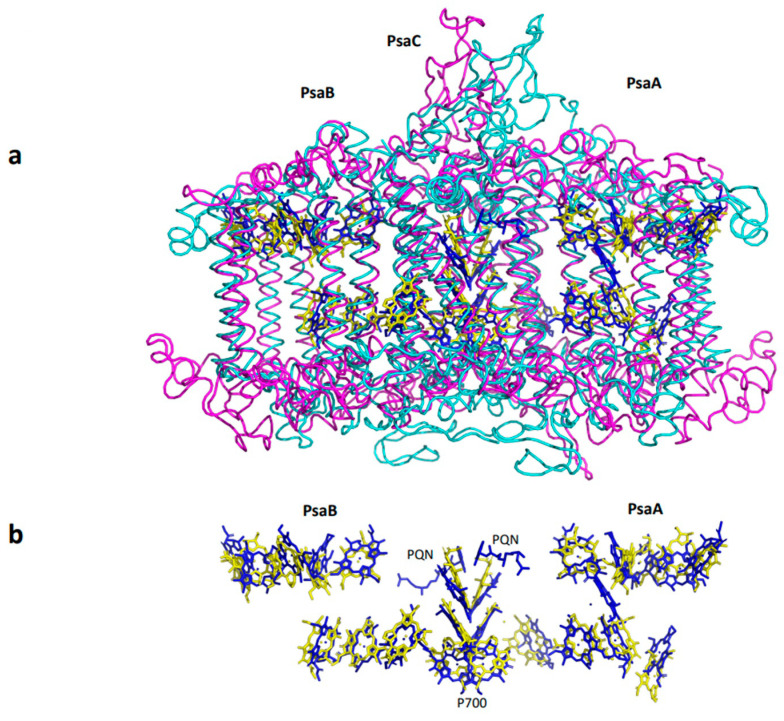
Superposition and comparison of the core subunits of *Chlorobium* Reaction Center with the core subunits of plant PSI. (**a**) Ribbon presentation of Pea plant PsaA, PsaB, and PsaC (PDB 5l8r—magenta) superposed on subunits A, a, and B of the *Chlorobium* reaction center (PDB 7z6q—cyan). A total of 23 chlorophyll molecules situated at identical positions are shown (blue—pea and yellow—*Chlorobium*). (**b**) The 23 chlorophyll molecules are situated at identical positions. The two quinone molecules are shown for orientation. All figures employed PyMOL for illustration.

**Figure 2 ijms-25-05073-f002:**
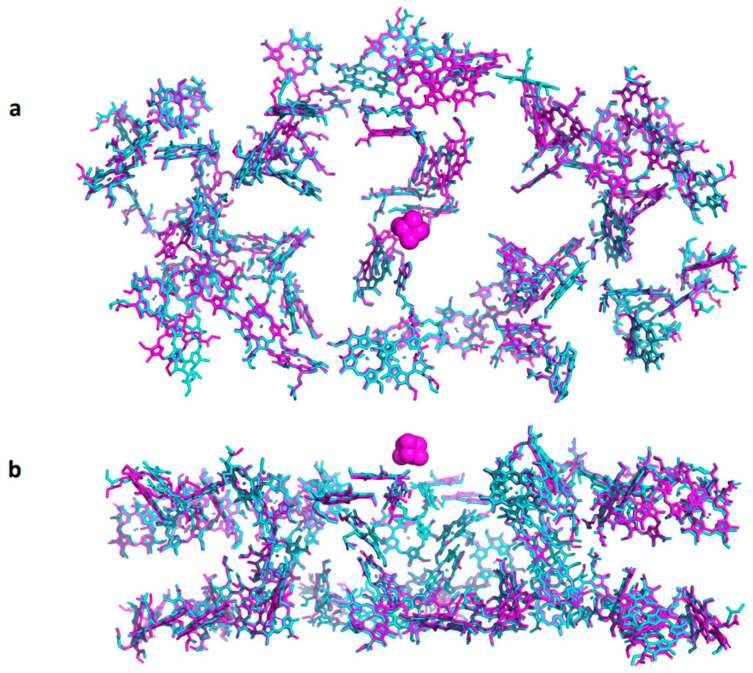
Eighty-four chlorophyll molecules share almost identical locations in PsaA and PsaB of cyanobacterial and higher plant PSI. The core subunits of Synechocystis (PDB 7z6q-cyan) and Pea plants (PDB 5l8r-magenta) were aligned and 4 chlorophylls out of 88 that did not share identical positions were removed. (**a**) View from the stroma. (**b**) View from the membrane.

**Figure 3 ijms-25-05073-f003:**
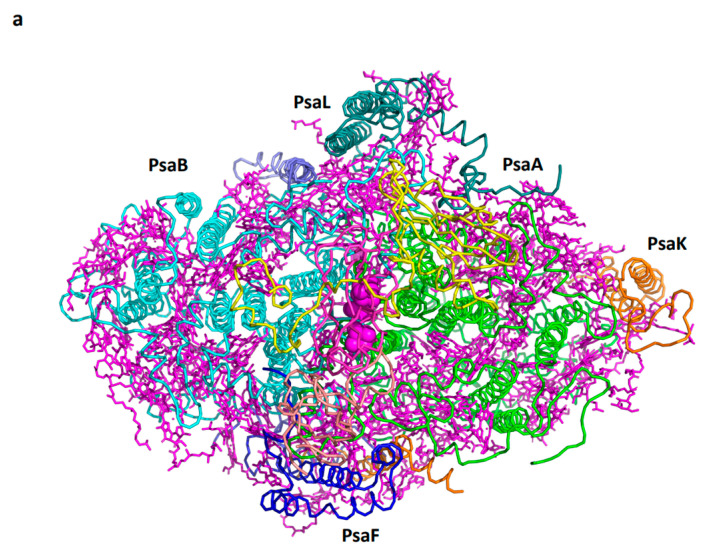
Conserved PSI core subunits. (**a**) A view from the stroma on the conserved core subunits. (**b**) Light-induced electron transfer pathway in PSI.

**Figure 4 ijms-25-05073-f004:**
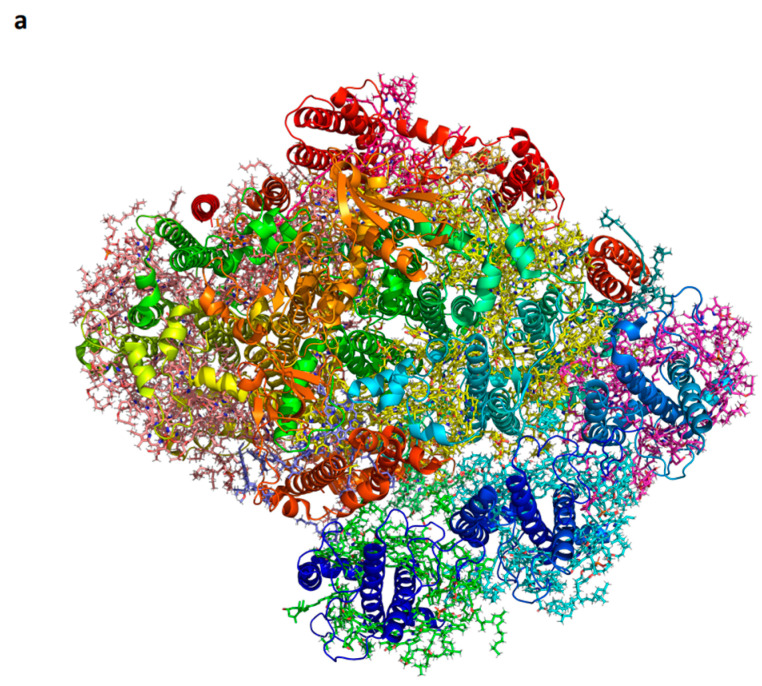
Comparison between PSI supercomplexes of the red algae *Cyanidioschyzon merolae* and the diatom *Chaetoceros gracilis*. (**a**) A view from the stroma on the cryo-EM structure of PSI from *C. merolae* (PDB 7blz). (**b**) A ribbon structure viewed from the stroma on the cryo-EM structure of PSI supercomplex from *C. gracilis* (PDB 6ly5). The common red lineage structure is in blue.

**Figure 5 ijms-25-05073-f005:**
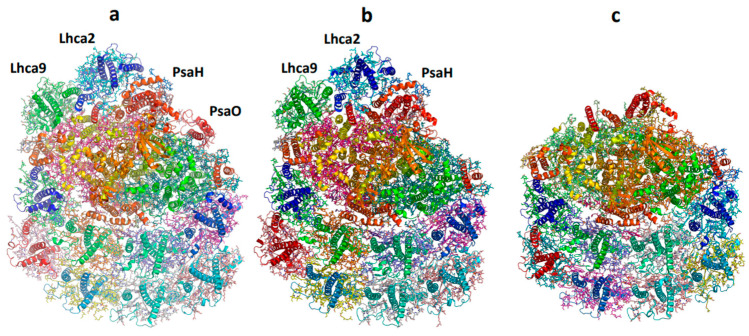
Effect of high light intensity on the subunit structure of *Chlorella ohadii*. (**a**) The structure of PSI from low light grown cells (PDB 6zzx). (**b**) The structure of PSI from high light-grown cells (PDB 6zzy). (**c**) A smaller conformer from high light-grown cells (PDB 7a4p).

**Figure 6 ijms-25-05073-f006:**
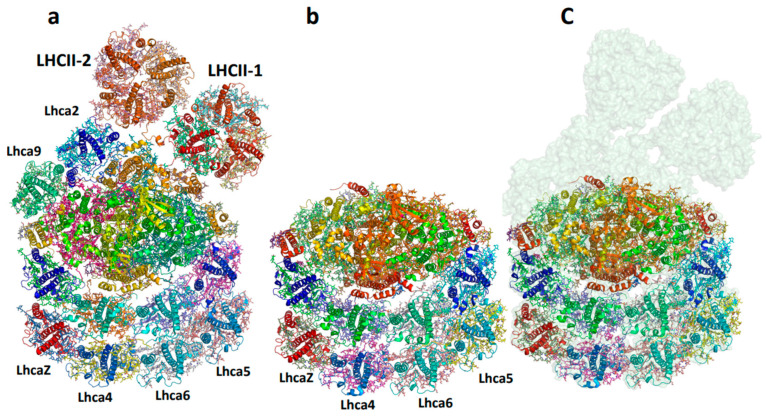
Structure of *C. reinhardtii* PSI isolated from b6/f temperature mutant grown at an elevated temperature. (**a**). The structure of state 2 PSI from optimally grown cells (PDB 7dz7). (**b**). PSI isolated from b6/f temperature mutant grown at elevated temperature (PDB 7r3k). (**c**). Structure of PSI isolated from b6/f temperature mutant grown et elevated temperature on the background of the surface presentation of the native complex at 80% transparency.

**Figure 7 ijms-25-05073-f007:**
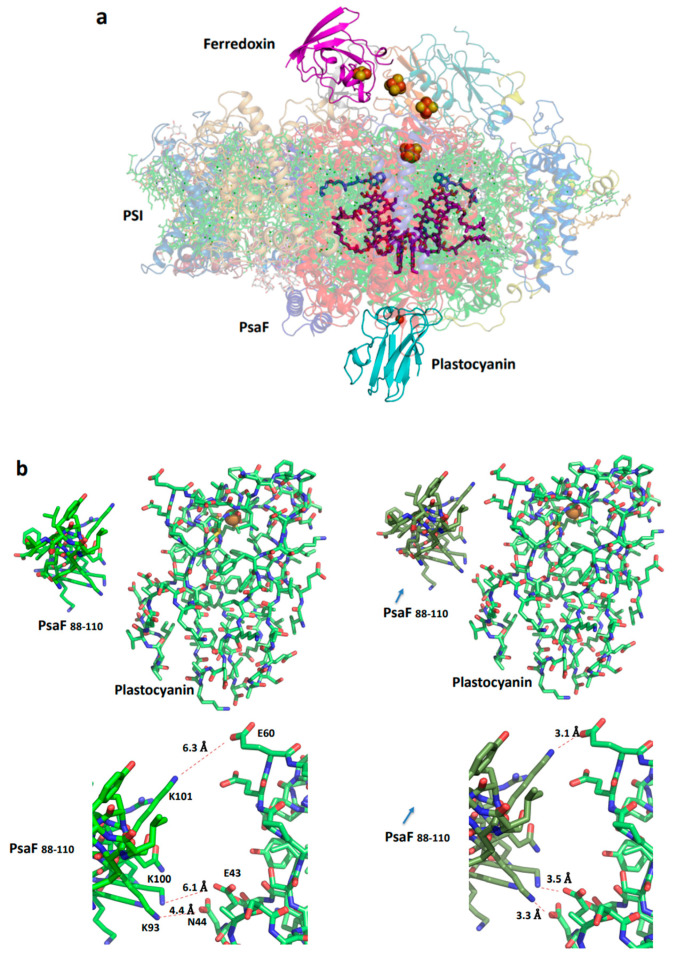
Structure of the plant plastocyanin-PSI-ferredoxin supercomplex. (**a**). A view from the membrane plane (PDB 6YEZ). The four iron-sulfur clusters are red and brown balls and the plastocyanin cupper is dark red. (**b**). Distances from the positively charged amino acids of PsaF and the negatively charged amino acids of plastocyanin (left). A suggested conformational change in PSI that moves toward bound oxidized plastocyanin. The arrow indicates the direction of the conformational change related to PsaF; three slat bridges that facilitate the dissociation of plastocyanin are formed. (**c**). The subunit junction in PSI is likely to be involved in mediating the conformational change.

**Figure 8 ijms-25-05073-f008:**
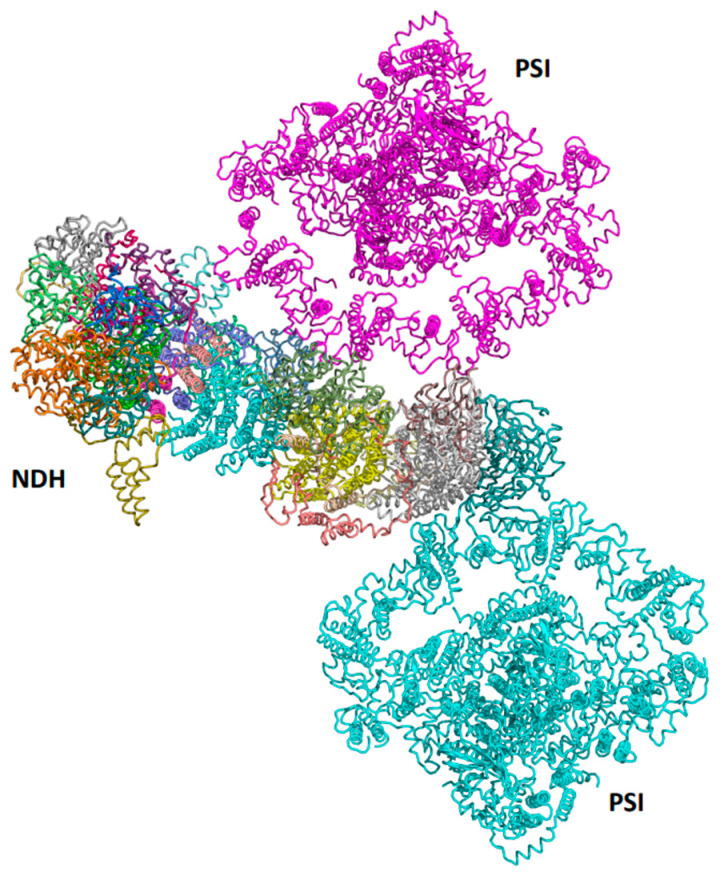
Structure of the plant PSI-NDH supercomplex. A view from the stroma on the supercomplex (PDB 7WG5) contains two PSI complexes bound to NDH through the LHCI belts.

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
