# Peer review of "Investigating the Balance between Structural Conservation and Functional Flexibility in Photosystem I"

_ijms, 2024, doi:10.3390/ijms25105073_

Round 1

Reviewer 1 Report

Comments and Suggestions for Authors

Comments on the Quality of English Language

Some places need to be corrected. Please see the comments attached.

Author Response

In this mini-review, the author described some new insights into the structure and function of PSI and its associated light-harvesting proteins, with a specific focus on the structural conservation of the core complex and variations of the peripheral light harvesting complexes. Recent results on the structural analysis of PSI-LHCI supercomplexes from various organisms and its related protein complexes are summarized, which offers a brief and timely overall view on what we have learned from recent studies and what are to be revealed in the future studies. This article is thus worth to be published, provided that the author takes the following into consideration in revising the manuscrcipt.

Lines 49-50: “Aligning the Chlorobium sequence with the cyanobacterial or higher plant sequences revealed a notably low identity, approximately 30% over 80 amino acids”. Which gene or genes does this sentence indicate?

Detailed in the citing reference

Lines 52-53: “Additionally, 23 chlorophyll molecules were found at almost identical locations (Figure 1b).”. However, green sulfur bacterial RC contains only 30 chlorophylls, and cyanobacterial and higher plant PSI contain 87-88 Chls. In other words, only 23 out of 87-88 Chls are conserved between green sulfur bacterial RC and PSI. This should be pointed out here, and the ref. reporting this should be cited (Chen et al., Architecture of the photosynthetic complex from a green sulfur bacterium. Science 370, eabb6350 (2020)).

Corrected

Legend to Fig. 2, last line: “where” should be corrected to “were”.

Corrected

Lines 82-84: “Even more puzzling is the short evolutionary period during which a random sequence of about 700 amino acids evolved to form PsaA and PsaB, presumably occurring over few hundred million years ...”. Previously, it has been stated that PsaA and PsaB is evolved from a single gene in green sulfur bacteria probably through gene duplication, so their evolution cannot be said “random”. What the “700 amino acids” represent? Also, it is not clear if the time is too short or not for the evolution of the duplicate gene PsaA and PsaB. The original text should be modified in relation to these contexts.

This refers to large photosynthetic polypeptide that is capable of light-induced “productive” charge separation.  

Lines 96-98: “Some remnants of these oceans were transported as meteorites containing primitive life forms, contributing to local panspermia and initiating life on Earth.” If this is the case, the “primitive life” should corresponds to “primitive green sulfur bacteria” or some similar life forms, and PSI still needs to be evolved from “primitive life”. In this context, would the “short evolutionary period” mentioned above be enough for this evolution?

Not necessary.

Legends to Fig. 3: Which organism does the structure represents? Please define. Line 142: “Variations in the Teem”. Should it be “Variations in the Theme”? Line 198: Ref. [62] is wrong here. It should be ref. 59 or 58?

Thanks! Corrected

Legends to Fig. 4: PDB code 7ly5 is wrong. Please clarify.

Thanks! Corrected to 6ly5

Line 250-251: Ref. [52] is cited here, but is it correct? Ref. [52] is crystal structural analysis, not cryo- EM structure. Please clarify.

Corrected

Where is ref. 71 cited?

Thanks! Corrected

Legends to Fig. 6: “frown et” should be “grown at”?

Corrected

Line 307: “Interaction between PSI with Electron Donors and Acceptors” may be “Interaction between PSI and Electron Donors and Acceptors”.

Corrected

Line 339, legends to Fig. 7: “casters” should be “clusters”.

Thanks! Corrected

Line 381, legends to Fig. 8: PDB code 7WG6 is the structure of an Neutral Omicron Spike Trimer, not the PSI-NDH supercomplex. Please clarify.

Thanks! Corrected to 7WG5

Reviewer 2 Report

Comments and Suggestions for Authors

1) Chapters must be distinguished in the manuscript. All subchapters in this publication cannot be part of a one chapter "Introduction". You may consider separating the Conclusion chapter.

2) The aim of this study is missing.

3) The selection of literature (96 items) is appropriate, but should be expanded. Many extensive fragments of the manuscript, which is a literature review, lack references to the literature, e.g. L 21-39, 46-61, 69-76, 92-102, etc.

4) The readability of Figures 1-8 should be improved.

5) Please indicate the author(s) of Figures 1-8.

6) L 85: "We propose ..." ? - this paper is single-author.

7) Latin names should be written in italics, e.g. Physcomitrella patens. (L 354, 357).

8) „References” were developed carelessly. The list of literature must be adapted to the journal's requirements.

Author Response

1) Chapters must be distinguished in the manuscript. All subchapters in this publication cannot be part of a one chapter "Introduction". You may consider separating the Conclusion chapter.

2) The aim of this study is missing.

This is a review

3) The selection of literature (96 items) is appropriate, but should be expanded. Many extensive fragments of the manuscript, which is a literature review, lack references to the literature, e.g. L 21-39, 46-61, 69-76, 92-102, etc.

4) The readability of Figures 1-8 should be improved.

5) Please indicate the author(s) of Figures 1-8.

All the figures were originally drowned according to the indicated PDBs

6) L 85: "We propose ..." ? - this paper is single-author.

Corrected

7) Latin names should be written in italics, e.g. Physcomitrella patens. (L 354, 357).

Corrected

8) „References” were developed carelessly. The list of literature must be adapted to the journal's requirements.

Please correct during edition.

Reviewer 3 Report

Comments and Suggestions for Authors

Dear Author,

The review manuscript "Investigating the Balance Between Structural Conservation and Functional Flexibility in Photosystem I" addresses a problem related to the structure of Photosystem I, which plays a very important function in the photosynthesis process. The manuscript contains interesting and valuable information obtained in the author's own previous research. Below is what should be improved in the manuscript.

- Abstract - The first part of the abstract (L: 8-13) was written without any reservations. Further on, I would suggest a more specific description referring to the content of the manuscript, including a summary of the content.

- Introduction - no cited literature items, it is not known on what sources the author prepared the description. This needs to be completed. I would suggest adding basic information about the photosynthesis process (a brief description of the process, the organisms in which it occurs, etc.) to introduce the topic of the manuscript. The introduction also lacks the purpose of the work and a reference to why this topic is important for science.

- I have reservations about the structure of the work, namely the order and numbering of subchapters. I would suggest that subsequent subsections begin with a different number (e.g. 2. Structure and functioning of PSI). At the beginning of this chapter, it would be advisable to describe the structure of PSI, and then the molecular architecture of primordial PSI core complex. - L: 46-62 no cited literature items on the basis of which the description was made. The same applies to text L: 92-102

- I would suggest writing impersonally instead of, for example, "I offer in line 14 and "we propose" in line 85, "we isolated" in line 122 (please correct it throughout the text of the work).

- Latin names of species should be written in italics, e.g. L: 50 (Chlorobium), 63, 65, 251, 277, 314, 351,

- species names written in English should start with lowercase letters, e.g. in line 123 oat (please correct them throughout the text)

- Figures - please provide the source of the figures (were they made using a specific program? specify what program. Figures are made professionally, I would suggest working on a better resolution

- The manuscript lacks a Conclusions chapter that would summarize the review

- References - are selected appropriately to the content of the manuscript. You should work on the aesthetics and adapt to the editorial requirements regarding the "References" chapter

Author Response

The review manuscript "Investigating the Balance Between Structural Conservation and Functional Flexibility in Photosystem I" addresses a problem related to the structure of Photosystem I, which plays a very important function in the photosynthesis process. The manuscript contains interesting and valuable information obtained in the author's own previous research. Below is what should be improved in the manuscript.

- Abstract - The first part of the abstract (L: 8-13) was written without any reservations. Further on, I would suggest a more specific description referring to the content of the manuscript, including a summary of the content.

- Introduction - no cited literature items, it is not known on what sources the author prepared the description. This needs to be completed. I would suggest adding basic information about the photosynthesis process (a brief description of the process, the organisms in which it occurs, etc.) to introduce the topic of the manuscript. The introduction also lacks the purpose of the work and a reference to why this topic is important for science.

The importance of this subject is self-evident.

- I have reservations about the structure of the work, namely the order and numbering of subchapters. I would suggest that subsequent subsections begin with a different number (e.g. 2. Structure and functioning of PSI). At the beginning of this chapter, it would be advisable to describe the structure of PSI, and then the molecular architecture of primordial PSI core complex. - L: 46-62 no cited literature items on the basis of which the description was made. The same applies to text L: 92-102

I prefer to leave it as is

- I would suggest writing impersonally instead of, for example, "I offer in line 14 and "we propose" in line 85, "we isolated" in line 122 (please correct it throughout the text of the work).

Corrected

- Latin names of species should be written in italics, e.g. L: 50 (Chlorobium), 63, 65, 251, 277, 314, 351,

Corrected

- species names written in English should start with lowercase letters, e.g. in line 123 oat (please correct them throughout the text)

Corrected

- Figures - please provide the source of the figures (were they made using a specific program? specify what program. Figures are made professionally, I would suggest working on a better resolution

All figures employed PyMOL for illustration.

- The manuscript lacks a Conclusions chapter that would summarize the review

- References - are selected appropriately to the content of the manuscript. You should work on the aesthetics and adapt to the editorial requirements regarding the "References" chapter

Round 2

Reviewer 2 Report

Comments and Suggestions for Authors

In my opinion, the manuscript in its current form (after the Author's correction) should not be published due to the significant deficiencies that still exist. The Author took into account only a few comments. However, he did not take into account the following comments:

- no new chapters were created in the manuscript; all the subsections of this manuscript are included in a single chapter, "Introduction",

- many of the extensive sections of the manuscript that constitute the literature review lack literature references. This is a review!!!

- the purpose of this study is missing,

- the Author(s) of Figures 1-8 are not given,

- "References" were developed carelessly. The Author's response "Please correct during edition" is surprising.

For this reason, I suggest rejecting the manuscript and giving the Author the opportunity to resubmit it.

Reviewer 3 Report

Comments and Suggestions for Authors

Dear Author,

As I suggested previously, the purpose of the work, rearranging the points of the subsections and Cocnlusions would significantly improve the quality of the manuscript. The author did not follow my suggestions. Therefore, I leave the decision to the Editors.